# Bilateral Global Nephrocalcinosis in a Uremic Puppy

**DOI:** 10.3390/vetsci11080338

**Published:** 2024-07-25

**Authors:** Maria Rizzo, Melissa Pennisi, Francesco Macrì, Annastella Falcone, Simona Di Pietro, Kamel Mhalhel, Elisabetta Giudice

**Affiliations:** 1Department of Veterinary Sciences, University of Messina, Polo Universitario Annunziata, 98168 Messina, Italy; rizzom@unime.it (M.R.); melpennisi@unime.it (M.P.); francesco.macri@unime.it (F.M.); kamel.mhalhel@unime.it (K.M.); egiudice@unime.it (E.G.); 2Veterinary Teaching Hospital, University of Messina, Polo Universitario Annunziata, 98168 Messina, Italy; simona.dipietro@unime.it

**Keywords:** dog, CKD, AKD, renal failure, renal biopsy, calcium-phosphorus product, intratubular calcification, B-mode ultrasound, color doppler

## Abstract

**Simple Summary:**

This study explores kidney disease in young dogs, focusing on early diagnosis, management, and the importance of staging for effective treatment. Highlighting mineral metabolism imbalances and complications like nephrocalcinosis, the study presents a case of severe renal failure with uremic syndrome and bilateral nephrocalcinosis in a 50-day-old puppy. Despite intensive care, the puppy’s condition worsened rapidly, leading to euthanasia. The study underscores the challenges in diagnosing and managing canine nephrocalcinosis in young animals. It emphasizes the need for further research to improve the understanding and treatment outcomes in such cases, enhancing the quality of life for animals suffering from this rare condition.

**Abstract:**

This study explores kidney disease in young dogs, focusing on early diagnosis, management, and the importance of staging for effective treatment. Highlighting mineral metabolism imbalances and complications such as nephrocalcinosis, the study presents a case of severe renal failure with uremic syndrome and bilateral nephrocalcinosis in a 50-day-old puppy. Despite intensive care, the puppy’s condition deteriorated rapidly, leading to euthanasia. The study underscores the challenges in diagnosing and managing canine nephrocalcinosis in young animals. It emphasizes the need for further research to improve the understanding and treatment outcomes in such cases, ultimately enhancing the quality of life for animals suffering from this rare condition.

## 1. Introduction

Chronic kidney disease (CKD), characterized by progressive worsening, is the most common pathology in older dogs and cats but is rare in young animals [1]. In contrast, acute kidney injury (AKI) encompasses a spectrum of conditions associated with sudden renal parenchymal damage [2,3]. In human medicine, AKI is defined as an abrupt decrease in kidney function within 7 days, whereas acute kidney disease (AKD) develops over 7–90 days, and CKD over more than 90 days [2]. Renal disease in immature or young adult dogs, not associated with primary renal inflammation, is defined as juvenile nephropathy. This includes a variety of pathologies such as agenesis, hypoplasia, dysplasia, glomerulopathies, tubulointerstitial nephropathies, and tubular transport dysfunction, described in different dog breeds [4]. When a familial basis has been detected, the renal disease is termed “familial nephropathy” [5]. Renal failure is the most feared consequence of both CKD and AKI/AKD, causing irreversible alterations in the body’s homeostasis, particularly in the balance of calcium and phosphorus [2,6]. Furthermore, alterations in mineral metabolism trigger a self-perpetuating cycle of kidney damage, accelerating the pathological process. The decreased number of functioning nephrons disrupts the homeostasis of elements, mainly phosphorus. Hyperphosphatemia triggers the development of secondary hyperparathyroidism, leading to an increase in the calcium-phosphorus product and tissue mineralization [7]. Phosphorus and calcium homeostasis are interlinked; both minerals are regulated by calciotropic hormones (parathyroid hormone/PTH, calcitonin, and calcitriol). An increase in plasma phosphorus concentration causes a reciprocal decrease in ionized calcium concentration. Calcitriol (1,25-dihydroxycholecalciferol), the active form of vitamin D, increases phosphorus and calcium absorption from the gastrointestinal tract. PTH increases phosphorus and calcium reabsorption from the bone and acts to increase calcium reabsorption and decrease phosphorus reabsorption from the glomerular filtrate in the renal tubules. An increased plasma phosphorus concentration stimulates increased PTH secretion and inhibits the formation of calcitriol in the kidney, creating homeostatic feedback loops. Increased plasma phosphorus concentration also inhibits the action of the 1-alpha-hydroxylase system, promoting soft tissue mineralization [7]. These disorders—mineral and bone disorders—are considered important complications of CKD, increasing the risk of soft tissue mineralization [8]. Moreover, hyperphosphatemia and the calcium-phosphorus product have been recognized as prognostic factors in dogs with CKD [9,10].

Today, an earlier diagnosis of chronic kidney disease is possible, allowing for therapeutic approaches that can block or minimize the rapid progression of the disease and improve the quality of life for affected animals. To provide adequate therapeutic support and patient monitoring, kidney disease staging should be performed [11]. Early diagnosis, staging, and substaging of the disease allow for a more accurate prognosis and guide decision-making regarding therapeutic management. Investigating complications that may accelerate kidney disease is also an important aspect of substaging CKD. This diagnostic approach has been created by the International Renal Interest Society (IRIS) [12].

Nephrocalcinosis is a heterotopic calcification characterized by the increased deposition of calcium salts (oxalate or phosphate) in the kidney [13]. It can be divided into two categories, intratubular or interstitial, with the renal papillae being the most affected [14]. Crystal attachment is an important step in the development of intratubular nephrocalcinosis. Crystals bind to and are taken up by renal epithelial cells, stimulating cell proliferation and inflammation. The mechanisms mediating crystal–cell attachment remain unclear, but there is likely an imbalance between the factors predisposing and inhibiting the deposition of calcium in kidney tissue [15]. Among the pathogenetic hypotheses on the origin of renal calcifications, the “Fixed particle model” best explains intratubular nephrocalcinosis [16,17]. Miyazawa et al. [18] showed that COX-2 and endogenous PGE2 regulate the attachment of calcium oxalate crystals to renal epithelial cells in humans. There are several diagnostic approaches for nephrocalcinosis. The most sensitive method compared to conventional radiography is ultrasound examination, although it is less specific than computed tomography [19]. Nevertheless, biopsy is the gold standard for confirming pathology. Histological examination allows for the identification of calcium salt deposit sites, namely the renal tubular cells, renal interstitium, and/or tubular lumen. In cases of lumen involvement, obstruction can cause tubulopathy that could evolve into chronic renal failure [20]. As stated above, nephrocalcinosis may be both the cause and the result of kidney disease and consequent renal failure.

Nephrocalcinosis is a feared complication in premature babies [19,21], but it is poorly described in puppies, even in the presence of congenital nephropathies [1]. Given the rarity of nephrocalcinosis in early life, this report aims to contribute to the knowledge of this condition to improve its clinical management.

## 2. Case Presentation

### 2.1. Clinical Findings

A 50-day-old, unvaccinated female mongrel puppy was evaluated for dysorexia, vomiting, polyuria, polydipsia, and profuse diarrhea. For approximately two weeks, the referring veterinarian treated her for suspected infectious gastroenteritis, but the clinical condition progressively worsened despite treatment with crystalloids, antiemetics, gastroprotectants, and antibiotics, resulting in severe renal failure.

On physical examination, the dog was depressed, cachectic (BCS: 2/9; 2.10 kg of body weight), and dehydrated (8%), with pale mucous membranes and prolonged capillary refill time (CRT: 2–3 s). The rectal temperature, heart rate, and respiratory rate were normal (38 °C, 100 beats/min, and 16 breaths/min, respectively). Thoracic auscultation was unremarkable, while an increase in borborygmi was noted on abdominal auscultation. Abdominal palpation revealed painless, hard, irregular kidneys and fluid-filled intestinal loops. No further abnormalities were detected.

### 2.2. Laboratory Findings

Hematology analysis, conducted using an automated analyzer (Procyte Dx, Idexx, Westbrook, MN, USA), revealed no significant abnormalities except for reticulocytopenia (7 K/µL; reference range: 10–110 K/µL) and monocytosis (2.14 K/µL; r.r: 0.16–2.12 K/µL). Biochemistry analysis, performed with an automated analyzer (Catalyst Dx, Idexx), showed severe alterations, consisting in high levels of serum nitrogen (BUN > 73.21 mmol/L; r.r. 1.17–4.83 mmol/L), creatinine (1502.83 µmol/L; r.r. 26.52–106.08 µmol/L), total calcium (2.64 mmol/L; r.r. 2.15–2.94 mmol/L), and phosphates (3.87 mmol/L; r.r. 1.65–3.36 mmol/L). The calcium to phosphate ratio was 0.88, while their product (Ca × P) was 127.9 mg^2^/dL^2^.

Serological exams for the common canine infectious diseases (distemper, hepatitis, parvovirosis and leptospirosis) tested negative.

Urinalysis showed a high urine protein (mg/dL) to creatinine (mg/dL) ratio (UPCR 3.14; r.r. < 0.5), hyposthenuria (1015; r.r. 1025–1035), glycosuria, cylindruria, and numerous kidney epithelial cells. No parasites were observed by the copromicroscopy exam.

### 2.3. Imaging Findings

Radiological examination was performed using an Analogic Radiographic/Fluoroscopic Table System (Dedalus Mb 90/20 IMX-2A, Imago Radiology S.r.l., Abbiategrasso (MI), Italy) with a digital radiography system (Fujifilm Medical Systems, Cernusco sul Naviglio, Italy). Two orthogonal projections (lateral and ventrodorsal) were performed. The x-ray exam showed a normal kidney size (3.5 × the length of the second lumbar vertebra) with increased radiopacity, especially in the cortex (Figure 1).

Ultrasound examination of both kidneys was performed on the dog while it was awake using the MyLab 40/Vet scanner (Esaote, Genova, Italy) equipped with linear (5.0- to 8.0-MHz) transducers. During B-mode ultrasonography, both kidneys were measured in the longitudinal plane and the maximum craniocaudal diameter was determined, showing a diameter of 45 mm in both kidneys. Both kidneys showed an increased medullary size and echogenicity with cortical thinning and increased echogenicity (Figure 2).

Based on the clinical, laboratory, radiological, and ultrasound findings, the diagnosis of severe renal failure with uremic syndrome and presumed bilateral nephrocalcinosis was issued. The puppy could be classified as having end-stage renal failure. The puppy was hospitalized and treated with intravenous crystalloids fluid therapy, antiemetics (maropitant, Cerenia, Zooetis Italia srl; 2 mg/kg/day subcutaneously), gastric secretion inhibitors (ranitidine, Zantadine, Ceva Vetem SpA; 2 mg/kg every 12 h subcutaneously), and protectors of the gastric mucosa (sucralfate, 0.5 g every 8 h per os). A critical care diet for kidney diseases (Renal liquid, Royal Canine) was also administered. After rehydration, fluids were set up to guarantee the water balance between outputs and inputs (“ins-and-outs”).

On the fourth day of hospitalization, despite the intensive care, the puppy’s clinical condition rapidly worsened, and the owner opted for euthanasia. The owner agreed to carry out further diagnostic tests, consisting of a color Doppler and a renal biopsy. He did not give the consent for a necropsy.

### 2.4. Doppler Examination

While the dog was awake, renal arterial and venous blood flows were visually examined using color Doppler ultrasonography. A pulsed-wave recording was performed on the interlobar arteries and veins. The vessels identified through color Doppler ultrasonography showed a normal vascular pattern (Figure 3).

### 2.5. Renal Biopsy and Histology

On the sedated puppy, an echo-assisted percutaneous renal biopsy from both kidneys was performed, followed by euthanasia as the owner had decided not to wake it.

The specimens have been maintained using routine procedures [22,23]. Tissue samples of fresh specimens were fixed in 4% paraformaldehyde in phosphate-buffered saline (PBS) (AAJ19943K2, Thermo Scientific, Waltham, MA, USA) 0.1 m (pH = 7.4) for 10 h, dehydrated through a graded ethanol series and then clarified in xylene for paraffin wax embedding. Embedded tissue samples were then cut into 5 µm thick serial sections and collected on gelatin-coated microscope slides. Sections were dried for 24 h, then processed for toluidine blue, hematoxylin and eosin, Masson’s trichrome with aniline blue (Bio-optica Milano S.p.A, Milan, Italy, CAT. #04–010802) staining according to the manufacturers’ instructions. The stained sections were observed under a Leica DMRB light microscope equipped with Leica MC 120 HD camera (Leica, Milan, Italy).

The histological examination allowed us to confirm the diagnostic suspicion of global bilateral nephrocalcinosis and to define the intratubular localization of calcium deposits. The lesions involved both renal parenchyma, mainly the proximal and distal cortical tubules and the cortico-medullary junction. In the tubulo-interstitial compartment, we recorded renal corpuscles (Gc) surrounded by a large pericapsular space (SPC) (Figure 4). These findings correspond to the characteristics of a typical lesion of renal dysplasia. This change is rare. The tubules showed microscopic alterations of different degrees. In some microscopic fields, tubules appeared dilated, filled with granular cylinders in the lumen, made up of protein material and mainly of cellular debris, with a structured epithelium. In other areas, there was an extensive pattern of intratubular calcified deposition with rarefaction and subsequent loss of the superficial epithelium, necrosis and, in some cases, tubulorexis. Calcified intratubular deposition was associated with alteration of surrounding epithelial cells, varying from flattening to evident signs of degeneration. The tubular lumen contained protein precipitates and cellular degradation products or concentric laminations of calcium deposit. All nephrons with intratubular calcium deposition also showed an evident suffering of the renal corpuscles (Figure 4).

## 3. Discussion and Conclusions

The underlying pathology of renal failure remains controversial. The rapid decline in renal function suggests AKD, while signs of chronicity, such as polyuria, polydipsia, weight loss, and pale mucous membranes, could indicate CKD [2]. It is certainly unusual to observe such severe kidney damage in a young puppy with no known toxin exposure. The occurrence of severe CKD in a puppy less than three months old is also rare; even rarer is the presentation of diffuse tubular nephrocalcinosis. Early diagnosis and targeted therapy can influence prognosis and improve quality of life, even in preterm neonates [19]. The most reliable and sensitive diagnostic tool is ultrasonography, which is useful for both screening and grading. It can distinguish between medullary, cortical, or global forms of nephrocalcinosis [19].

In the case in question, the involvement was bilateral and global, with greater severity at the cortical and cortico-medullary levels. The histological examination confirmed the diagnosis and allowed us to identify the tubulopathy with lumen involvement and obstruction, probably underlying the chronic renal failure.

Nephrocalcinosis is a morphological picture with multifactorial etiopathogenesis, but the urinary supersaturation and the following crystallization of renal parenchyma seem to be the most important aspects. Nephrocalcinosis can be linked to several causes, exogenous, hereditary, or to a mixture of other pathologies that do not always concern the kidney. In children, the main cause is represented by congenital tubulopathies (41% of cases), mainly distal tubular acidosis, followed by vitamin D intoxication (10%), hyperoxaluria (7%), idiopathic hypercalciuria (5%), and other rare causes (12%) [24,25]. In other studies, nephrocalcinosis was associated with hypercalciuria in 35% of cases, hereditary diseases in 23%, and idiopathic forms in 6% [26].

In this case report, it was not possible to detect the cause of severe nephropathy. On the case history, dietary and drug or exogenous toxic causes were excluded. There were no histological aspects of disorganized nephrogenesis, which could result from insults experienced during the fetal or neonatal period, nor of primary renal or urinary tract inflammation. Furthermore, the presence of amorphous material in the renal parenchyma was also not observed, such as in genetic collagen–fibrotic glomerulopathies described in some litters [1]. Nephron’s damage and a reduction in glomerular filtration rate (GFR) affect the homeostasis of several solutes primarily excreted in the urine, including phosphorus. Phosphorus deposition occurs in soft tissues and mineralization ensues, a phenomenon that could contribute to progressive kidney injury, with kidney disease evolution and extrarenal (mainly cardiovascular) effects. Like in humans, the progression of kidney damage is induced by the accumulation of phosphate in the body resulting in renal calcification [27]. According to IRIS, plasmatic phosphorus levels are essential to stage renal disease [2,3]. Hyperphosphatemia is a prognostic factor for shorter survival time in dogs with CKD [28,29]. Recently, Miyakawa et al. [27] reported that increased serum fibroblast growth factor (FGF)-23 concentration was a significant risk factor for the subsequent development of hyperphosphatemia in normophosphatemic dogs with CKD, and it can be used as a tool to prevent the development of hyperphosphatemia.

In this puppy, the serum electrolytes were imbalanced with the calcium/phosphorus ratio thus supporting the hypothesis of hyperparathyroidism and its possible association with renal failure.

The assumption that ectopic calcification will occur when the so-called calcium-phosphorus product (Ca × P) exceeds a particular threshold has become standard practice in nephrology [30]. As the disease progresses to a terminal stage, the calcium-phosphorus product increases, thereby raising the likelihood of ectopic calcifications. The calcium-phosphorus product values that are considered to be at risk of soft tissue mineralization are higher than 60–70 mg^2^/dL^2^ [31]. In young animals, however, growth requirements (bones, in particular) lead to increases in plasma levels of Ca and P. In this period, even if the Ca × P product exceeds that limit, the skeletal demands generally ensure the prevention of soft tissue mineralization. In a recent study, calcium and phosphorus levels in growing healthy dogs (43–53 days old) were investigated and the average Ca × P product of 115 mg^2^/dL^2^ was measured [32]. The product value in our puppy was much higher (127.9 mg^2^/dL^2^) and may have promoted metastatic calcium deposition. Given the rarity of kidney disease in growing animals, however, we have no value to refer to as a causal factor.

In CKD and dialysis patients, preventive therapy is now an established protocol. Low-dose calcitriol treatment has beneficial effects on the regulation of parathyroid hormone and calcium levels, and direct nephroprotective effects on renal tubular regeneration are observed. In human nephrology, it is common practice to use a combined therapy with calcitriol, or vitamin D analogues, in conjunction with inhibitors of the renin–angiotensin–aldosterone system, dietary phosphate restriction, and calcimimetics [30,33]. Some studies have reported that therapeutic diets could affect mineral metabolism in dogs with CKD [28,34].

In this case report, after a 2-week clinical course and 4-day intensive care hospitalization, the puppy was euthanized. Regardless of the primary cause of the kidney damage, the early onset, severity, and rapid progression of the illness led to a poor outcome. With the clinical data at our disposal, it was not possible to arrive at a causal diagnosis; we could only localize the damage. However, congenital nephropathy remains a probable hypothesis. A slower and less severe clinical course, along with greater financial resources, could have allowed for more accurate ancillary examinations, such as blood gas analysis, electrolyte panels, endocrine profiles, immunohistochemistry, electron microscopy, and genetic analysis, to reach a causative diagnosis.

Despite these limitations, given the scarcity of references on canine nephrocalcinosis, the described clinical case can contribute to the study of this condition, which is even rarer in puppies. Further studies are needed to investigate the factors that can anticipate the development of nephrocalcinosis, allowing for early conservative measures to delay the progression of renal impairment and improve the quality and length of life in nephropathic patients.

## Figures and Tables

**Figure 1 vetsci-11-00338-f001:**
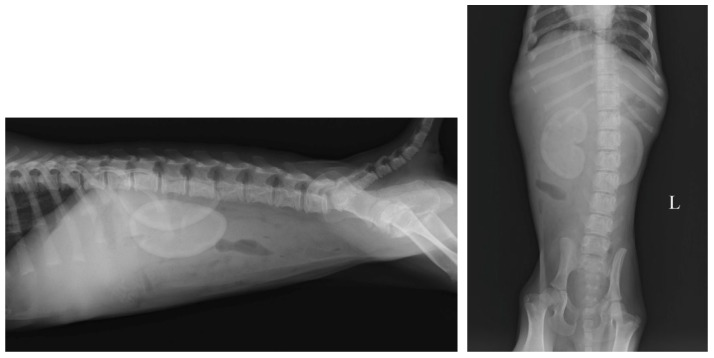
Lateral and ventrodorsal radiographs of the abdomen: note both kidneys with increased radiopacity. (“L” indicate left side).

**Figure 2 vetsci-11-00338-f002:**
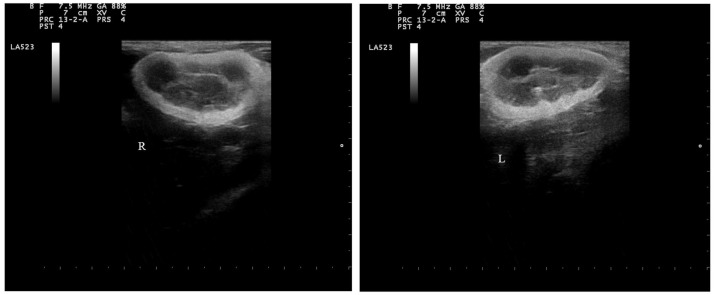
B-mode ultrasonography of kidneys. Note the increased medullary size and cortical thinning in both kidneys. (“R” and “L” indicate right and left kidney, respectively).

**Figure 3 vetsci-11-00338-f003:**
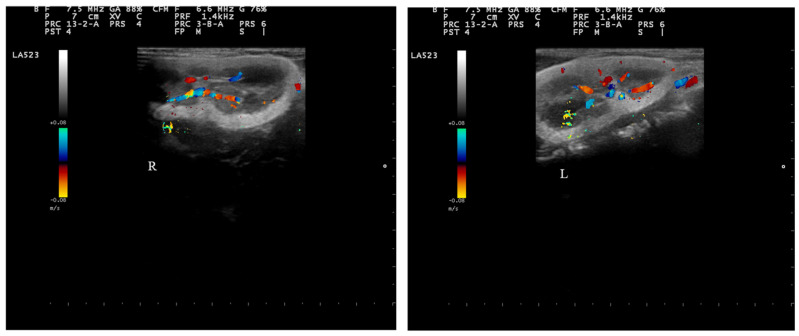
Color flow Doppler sonography in the renal vessels. (“R” and “L” indicate right and left kidney, respectively).

**Figure 4 vetsci-11-00338-f004:**
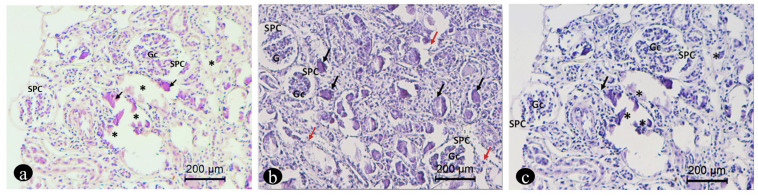
Photomicrographs of the renal biopsy sections stained with hematoxylin–eosin (**a**) and toluidine blue stain (**b**,**c**) from the renal biopsy section. The photomicrographs show the tubulo-interstitial compartment with renal corpuscles (Gc) surrounded by a large pericapsular space (SPC). The tubular lumen contained protein precipitates and cellular degradation products or concentric laminations of calcium deposit (black arrows). Obvious signs of cellular degeneration (red arrows). Severe regressive changes in the tubular epithelium with necrosis (asterisk).

## Data Availability

The data presented in this study are available on request from the corresponding author.

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
