# Peer review of "Bilateral Global Nephrocalcinosis in a Uremic Puppy"

_vetsci, 2024, doi:10.3390/vetsci11080338_

Round 1

Reviewer 1 Report (New Reviewer)

Comments and Suggestions for Authors

This case report entitled "Bilateral global nephrocalcinosis in a uremic puppy" contains useful information, especially in the differential diagnosis of renal failure in young dogs. While recognizing the clinical value of this report, I have a few comments.

Clinicopathologic data are lacking for an accurate diagnosis. To evaluate the cause and pathogenesis of tissue calcification, ionized calcium, and PTH need to be evaluated, and this is considered a major limitation of this report. The results of plasma electrolyte (Na/K/Cl) concentrations and blood gas analysis should also be included in the text.

The authors treated the case as end-stage chronic kidney disease (IRIS stage 4 CKD). However, it should be appropriate to treat this dog as having acute kidney injury (IRIS stage 5). Even if CKD is considered the underlying disease, a sudden decline in renal function should be treated as AKI. The IRIS stage for CKD is only used in stable patients.

An echo-guided percutaneous renal biopsy was performed immediately before euthanasia. Why was it not done immediately after euthanasia? I understand why a necropsy couldn't be done, but I didn't understand why the sample was taken before euthanasia.

Histopathology; the method of tissue preparation should be briefly described. Formalin fixation, paraffin embedding? What stains were used for renal histopathology? For general observation, not only hematoxylin-eosin (HE) but also other stains (periodic acid-Schiff (PAS), Masson's trichrome, etc.) are needed for renal pathology. Von Kossa's stain is required to access the localization of calcified deposits.

Fig. 4: Toluidine blue stain is not a general stain for renal pathology. The photographs taken from HE (or PAS) are needed. In addition, that taken from Von Kossa stains necessity in this case.

Were there the findings of renal dysplasia? This must be one of the most common kidney diseases in puppies with renal failure. Immature glomeruli and tubules appeared to be present in Figure 4, although it is difficult to distinguish with toluidine blue staining.

Author Response

Dear Editor,

Thank you very much for reviewing the manuscript titled "Bilateral Global Nephrocalcinosis in a Uremic Puppy" (Manuscript ID: vetsci-3048437).

We are pleased to learn that the reviewer recommended our work for publication. We have diligently addressed the majority of the reviewer's concerns, as detailed below, and have made corresponding modifications to the manuscript. We trust that the revisions we have made will render the manuscript suitable for publication.

We sincerely appreciate your assistance and valuable suggestions throughout this process. We eagerly await your feedback on our final version.

We have carefully considered the reviewer's new suggestions. All responses to the specific comments are reported below. Changes made to the manuscript are highlighted in red.

Thank you once again.

Review Report 1

Open Review

Quality of English Language

(x) I am not qualified to assess the quality of English in this paper
( ) English very difficult to understand/incomprehensible
( ) Extensive editing of English language required
( ) Moderate editing of English language required
( ) Minor editing of English language required
( ) English language fine. No issues detected

Yes

Can be improved

Must be improved

Not applicable

Does the introduction provide sufficient background and include all relevant references?

(x)

( )

( )

( )

Is the research design appropriate?

( )

(x)

( )

( )

Are the methods adequately described?

( )

( )

(x)

( )

Are the results clearly presented?

( )

(x)

( )

( )

Are the conclusions supported by the results?

( )

(x)

( )

( )

Comments and Suggestions for Authors

This case report entitled "Bilateral global nephrocalcinosis in a uremic puppy" contains useful information, especially in the differential diagnosis of renal failure in young dogs. While recognizing the clinical value of this report, I have a few comments.

Clinicopathologic data are lacking for an accurate diagnosis. To evaluate the cause and pathogenesis of tissue calcification, ionized calcium, and PTH need to be evaluated, and this is considered a major limitation of this report. The results of plasma electrolyte (Na/K/Cl) concentrations and blood gas analysis should also be included in the text.

This is one of the limitations of the work due to financial constraints.

The authors treated the case as end-stage chronic kidney disease (IRIS stage 4 CKD). However, it should be appropriate to treat this dog as having acute kidney injury (IRIS stage 5). Even if CKD is considered the underlying disease, a sudden decline in renal function should be treated as AKI. The IRIS stage for CKD is only used in stable patients.

We agree with the suggestion. We have added references to ARF in the text.

An echo-guided percutaneous renal biopsy was performed immediately before euthanasia. Why was it not done immediately after euthanasia? I understand why a necropsy couldn't be done, but I didn't understand why the sample was taken before euthanasia.

Because the owner decided to proceed with euthanasia during the procedure, it was determined not to wake the puppy up after the biopsy.

Histopathology; the method of tissue preparation should be briefly described. Formalin fixation, paraffin embedding? What stains were used for renal histopathology? For general observation, not only hematoxylin-eosin (HE) but also other stains (periodic acid-Schiff (PAS), Masson's trichrome, etc.) are needed for renal pathology. Von Kossa's stain is required to access the localization of calcified deposits.

Thank you for highlighting this issue, we have added a brief description of the tissue preparation methods used in our study as requested.

Fig. 4: Toluidine blue stain is not a general stain for renal pathology. The photographs taken from HE (or PAS) are needed. In addition, that taken from Von Kossa stains necessity in this case.

We have modified the figure as requested. We have added hematoxylin-eosin and Masson's trichrome with aniline stained section photomicrographs to the manuscript.

Were there the findings of renal dysplasia? This must be one of the most common kidney diseases in puppies with renal failure. Immature glomeruli and tubules appeared to be present in Figure 4, although it is difficult to distinguish with toluidine blue staining.

We have reported the histological characteristics of renal dysplasia in the Masson's trichrome with aniline stained section’ photomicrographs.

Reviewer 2 Report (New Reviewer)

Comments and Suggestions for Authors

General: Overall, I think this is an interesting case study. It is certainly unusual to see such severe kidney damage in such a young puppy with no known toxin exposure.

One point I think must be addressed is the use of the term "chronic" kidney disease. By definition, chronic kidney disease occurs when functional or structural changes occur to the kidney for 3 or more months. While there is no doubt this is a case of severe renal damage, can we truly call it chronic kidney disease in a puppy that has not reached 3 months of age?

Minor comments:

Line 72: Consider starting a new paragraph here to discuss nephrocalcinosis

Lines 91-92: The significance of this statement as its own paragraph is unclear to this reviewer. Please provide further explanation or consider moving it and combining it with information elsewhere to give it more context. For instance, it seems to this reviewer that these lines would fit better in the paragraph above the one discussing diagnosis of nephrocalcinosis.

Physical examination section: This is a minor point, but I am assuming lymph nodes were within normal limits? Please also include this information if available.

Biochemistry results: This comment comes mostly from curiosity…were sodium, potassium, and chloride measured as well? It would be interesting to see how altered those values may have been with such severe kidney disease.

Comments on the Quality of English Language

A few sentences could be improved upon (lines 236-237), but overall the English was fine. 

Author Response

Dear Editor,

Thank you very much for reviewing the manuscript titled "Bilateral Global Nephrocalcinosis in a Uremic Puppy" (Manuscript ID: vetsci-3048437).

We are pleased to learn that the reviewer recommended our work for publication. We have diligently addressed the majority of the reviewer's concerns, as detailed below, and have made corresponding modifications to the manuscript. We trust that the revisions we have made will render the manuscript suitable for publication.

We sincerely appreciate your assistance and valuable suggestions throughout this process. We eagerly await your feedback on our final version.

We have carefully considered the reviewer's new suggestions. All responses to the specific comments are reported below. Changes made to the manuscript are highlighted in red.

Thank you once again.

Review Report 2

Open Review

Quality of English Language

( ) I am not qualified to assess the quality of English in this paper
( ) English very difficult to understand/incomprehensible
( ) Extensive editing of English language required
( ) Moderate editing of English language required
(x) Minor editing of English language required
( ) English language fine. No issues detected

Yes

Can be improved

Must be improved

Not applicable

Does the introduction provide sufficient background and include all relevant references?

(x)

( )

( )

( )

Is the research design appropriate?

( )

( )

( )

(x)

Are the methods adequately described?

(x)

( )

( )

( )

Are the results clearly presented?

(x)

( )

( )

( )

Are the conclusions supported by the results?

( )

(x)

( )

( )

Comments and Suggestions for Authors

General: Overall, I think this is an interesting case study. It is certainly unusual to see such severe kidney damage in such a young puppy with no known toxin exposure.

One point I think must be addressed is the use of the term "chronic" kidney disease. By definition, chronic kidney disease occurs when functional or structural changes occur to the kidney for 3 or more months. While there is no doubt this is a case of severe renal damage, can we truly call it chronic kidney disease in a puppy that has not reached 3 months of age?

Thank you for your input. We have revised the text accordingly. We have added the following clarification: “In human medicine, acute kidney injury (AKI) is defined as an abrupt decrease in kidney function occurring in less than 7 days, acute kidney disease (AKD) develops over 7–90 days, and chronic kidney disease (CKD) is defined by changes occurring for more than 90 days.” This adjustment ensures accurate terminology is used in describing the kidney condition in the puppy.

Minor comments:

Line 72: Consider starting a new paragraph here to discuss nephrocalcinosis

Thank you for your input. We have revised the text accordingly.

Lines 91-92: The significance of this statement as its own paragraph is unclear to this reviewer. Please provide further explanation or consider moving it and combining it with information elsewhere to give it more context. For instance, it seems to this reviewer that these lines would fit better in the paragraph above the one discussing diagnosis of nephrocalcinosis.

Thank you for your comment. We have revised the text accordingly.

Physical examination section: This is a minor point, but I am assuming lymph nodes were within normal limits? Please also include this information if available.

The lymph nodes were within normal limits, and we included only the altered clinical data in the text. We have added this sentence: No further anomaly was detected.

Biochemistry results: This comment comes mostly from curiosity…were sodium, potassium, and chloride measured as well? It would be interesting to see how altered those values may have been with such severe kidney disease.

Unfortunately, no further data is available due to the owner's financial constraints.

Comments on the Quality of English Language

A few sentences could be improved upon (lines 236-237), but overall the English was fine. 

We improved the sentence.

Reviewer 3 Report (New Reviewer)

Comments and Suggestions for Authors

Chronic renal failure is a very important problem in veterinary medicine. As the authors rightly point out in the introduction, it mainly affects older animals, but unfortunately it sometimes occurs in young animals and its course can be fatal for the patient. The introduction is well written and informative.
The authors describe the case of a very young puppy at 50 days of age.
2.2- please indicate which apparatus was used for the blood test
2.2. please indicate the results on the SI scale, e.g. creatinine level umol/l
2.5. was the histology stained using H-E staining - please specify
Discussion is appropriate.
Literature is appropriate

Author Response

Dear Editor,

Thank you very much for reviewing the manuscript titled "Bilateral Global Nephrocalcinosis in a Uremic Puppy" (Manuscript ID: vetsci-3048437).

We are pleased to learn that the reviewer recommended our work for publication. We have diligently addressed the majority of the reviewer's concerns, as detailed below, and have made corresponding modifications to the manuscript. We trust that the revisions we have made will render the manuscript suitable for publication.

We sincerely appreciate your assistance and valuable suggestions throughout this process. We eagerly await your feedback on our final version.

We have carefully considered the reviewer's new suggestions. All responses to the specific comments are reported below. Changes made to the manuscript are highlighted in red.

Thank you once again.

Review Report 3

Open Review

Quality of English Language

(x) I am not qualified to assess the quality of English in this paper
( ) English very difficult to understand/incomprehensible
( ) Extensive editing of English language required
( ) Moderate editing of English language required
( ) Minor editing of English language required
( ) English language fine. No issues detected

Yes

Can be improved

Must be improved

Not applicable

Does the introduction provide sufficient background and include all relevant references?

( )

(x)

( )

( )

Is the research design appropriate?

(x)

( )

( )

( )

Are the methods adequately described?

(x)

( )

( )

( )

Are the results clearly presented?

(x)

( )

( )

( )

Are the conclusions supported by the results?

(x)

( )

( )

( )

Comments and Suggestions for Authors

Chronic renal failure is a very important problem in veterinary medicine. As the authors rightly point out in the introduction, it mainly affects older animals, but unfortunately it sometimes occurs in young animals and its course can be fatal for the patient. The introduction is well written and informative.
The authors describe the case of a very young puppy at 50 days of age.

2.2. please indicate which apparatus was used for the blood test

Thank you for your suggestion. We have revised the text accordingly.

2.2. please indicate the results on the SI scale, e.g. creatinine level umol/l

Thank you for your input. We have revised the text accordingly.

2.5. was the histology stained using H-E staining - please specify

The histology photomicrographs were stained with toluidine blu, still we have added both hematoxylin-eosin and Masson's trichrome with aniline stained section’ photomicrographs.

Discussion is appropriate.
Literature is appropriate.

Round 2

Reviewer 1 Report (New Reviewer)

Comments and Suggestions for Authors

Dear authors,

I have yet some minor comments, as follows.

Line 12: remove unnecessary parentheses

Lin2 149: end-stage kidney disease (IRIS stage 5 AKD or 4 CKD); This statement is inappropriate. There is no established definition of AKD in dogs and cats; for AKI, it would be IRIS grade 5 (not a stage); CKD stages and AKI grades cannot be defined as the same condition. In the present case, there is no sense in forcing a stage or grade classification, and the description of end-stage renal failure or severe renal failure is sufficient.

Figure 2: I have confirmed that you have done Masson's trichrome and HE stains for histopathological analysis. Generally, von Kossa staining is required to demonstrate calcium deposition in the tissue section, but in this case, there is no doubt of calcification from the radiographic and echographic findings. However, the quality of the Masson's trichrome stain in Figure 2 is quite poor; the quality of the HE stain is also poor but within a possible acceptable range. If the quality of the Masson's trichrome stain could not be improved, it should be removed from Figure 2.

Author Response

Thank you very much for reviewing the manuscript titled "Bilateral Global Nephrocalcinosis in a Uremic Puppy" (Manuscript ID: vetsci-3048437).

We are pleased to learn that the reviewer recommended our work for publication. We have diligently addressed the majority of the reviewer's concerns, as detailed below, and have made corresponding modifications to the manuscript. We trust that the revisions we have made will render the manuscript suitable for publication.

We sincerely appreciate your assistance and valuable suggestions throughout this process. We eagerly await your feedback on our final version.

We have carefully considered the reviewer's new suggestions. All responses to the specific comments are reported below. Changes made to the manuscript are highlighted in red.

Thank you once again.

Review Report Form 1

Open Review

Quality of English Language

(x) I am not qualified to assess the quality of English in this paper
( ) English very difficult to understand/incomprehensible
( ) Extensive editing of English language required
( ) Moderate editing of English language required
( ) Minor editing of English language required
( ) English language fine. No issues detected

Yes

Can be improved

Must be improved

Not applicable

Does the introduction provide sufficient background and include all relevant references?

(x)

( )

( )

( )

Is the research design appropriate?

( )

(x)

( )

( )

Are the methods adequately described?

( )

(x)

( )

( )

Are the results clearly presented?

( )

( )

(x)

( )

Are the conclusions supported by the results?

(x)

( )

( )

( )

Comments and Suggestions for Authors

Dear authors,

I have yet some minor comments, as follows.

Thank you for your comments. I have made the requested changes as follows:

Line 12: remove unnecessary parentheses

Thank you for your suggestion. The parentheses were a typographical error. We have revised the text accordingly.

Line 149: end-stage kidney disease (IRIS stage 5 AKD or 4 CKD); This statement is inappropriate. There is no established definition of AKD in dogs and cats; for AKI, it would be IRIS grade 5 (not a stage); CKD stages and AKI grades cannot be defined as the same condition. In the present case, there is no sense in forcing a stage or grade classification, and the description of end-stage renal failure or severe renal failure is sufficient.

Thank you for highlighting this issue, we revised the text accordingly. Now, it simply describes end-stage renal failure without forcing a stage or grade classification: “The puppy could be classified as end-stage renal failure.”

Figure 2: I have confirmed that you have done Masson's trichrome and HE stains for histopathological analysis. Generally, von Kossa staining is required to demonstrate calcium deposition in the tissue section, but in this case, there is no doubt of calcification from the radiographic and echographic findings. However, the quality of the Masson's trichrome stain in Figure 2 is quite poor; the quality of the HE stain is also poor but within a possible acceptable range. If the quality of the Masson's trichrome stain could not be improved, it should be removed from Figure 2.

Thank you for your suggestion. Since the quality of the Masson's trichrome stain (Figure 4b) could not be improved, it has been removed from Figure 4.

This manuscript is a resubmission of an earlier submission. The following is a list of the peer review reports and author responses from that submission.

Round 1

Reviewer 1 Report

Comments and Suggestions for Authors

Dear authors,

Introduction

Lines 71 to 75 - Reference is due

Case presentation

Lines 101 to 102 - Calcium to phosphate ratio

Line 105 - Urine Protein to Creatinine Ratio (uPCR 3.14 units -mg/mmol (?); r.r. <0.5),

Line 131- How can you justify the use of this antiemetic when its use is only safe after 8 weeks of age?

Line 211 - Please harmonize language "calcium/phosphorus ratio" instead "Calcium/ phosphate ratio"

Line 233 - Calcimimetics

Author Response

Dear reviewer, 

thank you for your suggestions. 

We modified the manuscript as suggested. 

We hope that now the manuscript is ready to be published. 

Best regards 

Simona Di Pietro

Reviewer 2 Report

Comments and Suggestions for Authors

The manuscript presents an interesting clinical case.

However, a very early diagnostic hypothesis is made with a lack of clinical data. I think the introduction should be expanded. I think that there is a lack of data to know the cause of the disease, which would be the really important thing.

The data should be better discussed and expanded for the correct diagnosis.

I believe it is necessary to make a better and more in-depth presentation for the clinical validity of the clinical case presented.

Author Response

Dear reviewer, 

thanks for your suggestions. 

We modified the manuscript as suggested. 

We hope that now the manuscript is ready to be published. 

Best regards 

Simona Di Pietro

Round 2

Reviewer 2 Report

Comments and Suggestions for Authors In my opinion the manuscript has been insufficiently modified and I believe that it should not be published.